# Vadose Zone Lag Time Effect on Groundwater Drought in a Temperate Climate

**Buruk Kitachew Wossenyeleh [1],*** , **Boud Verbeiren [2]** , **Jan Diels [1]** and **Marijke Huysmans [1,2]**

[1] Department of Earth and Environmental Sciences, KU Leuven, Celestijnenlaan 200E,
BE-3001 Leuven, Belgium; Jan.Diels@kuleuven.be (J.D.); marijke.huysmans@kuleuven.be or
marijke.huysmans@vub.be (M.H.)

[2] Department of Hydrology and Hydraulic Engineering, Vrije Universiteit Brussel, Pleinlaan 2,
BE-1050 Brussels, Belgium; Boud.Verbeiren@vub.be

* Correspondence: burukkitachew.wossenyeleh@kuleuven.be or bkwdae@gmail.com

**Abstract:** An essential factor in the propagation of drought, from meteorological drought to groundwater drought, is the delay between a precipitation event and the groundwater recharge reaching the groundwater table. This delay, which mainly occurs in the vadose zone of the hydrological cycle, is often poorly studied. Therefore, this paper proposes a method for estimating the spatially distributed delay in the vadose zone using the kinematic wave approximation of Richards' equation combined with the van Genuchten–Burdine and Brooks–Corey parametric model. The modeling was approached (1) using a detailed parametrization of soil and geological layers and (2) using lumped hydraulic and physical properties of geological layers. The results of both approaches were compared against the physically based flow model Hydrus-1D. This analysis shows that using a detailed parametrization of soil and geological layers results in good comparison, with a Nash–Sutcliffe efficiency of 0.89 for Brooks–Corey and 0.80 for van Genuchten–Burdine. The delay result of the Brooks–Corey model was incorporated into the groundwater recharge time series from 1980 to 2013 to analyze the effect of this delay on groundwater drought. The results show that the delay in the vadose zone influences groundwater drought characterization features such as the number, duration, and intensity of drought events.

**Keywords:** groundwater drought; groundwater recharge delay; vadose zone; kinematic wave approximation; drought propagation

---

## 1. Introduction

Drought can be described as a temporary decrease in water availability over a significant period of time. Drought is a direct result of deficient precipitation and, as such, it is mainly a meteorological-related hazard. Typically, three definitions of drought are used: meteorological, agricultural, and hydrological drought. The first refers to a period with little or no precipitation; the second refers to a shortage of water in the soil and as such for vegetation; the third describes the impact on hydrological water bodies. It can affect both surface and groundwater resources. Where this concerns groundwater bodies, the term groundwater drought is used. Groundwater drought can be defined as a temporary decrease in groundwater availability over a significant period of time. This drought causes decreased groundwater levels and discharge to the surface water system [1].

In Belgium, Tricot et al. [2] have shown that drought periods have not intensified during the last century. The drought periods were defined as the number of consecutive days without significant precipitation (less than 0.5 mm) for the six hottest months of the year. However, the study area experienced long periods of drought and low groundwater levels in the past four years. According to

Verbeiren et al. [3], the main influencing factors for groundwater drought are climate, land use/land cover (LULC), and groundwater demand for human activities.

An essential factor in the propagation of drought, from meteorological drought to groundwater drought, is the delay between a precipitation event and the resulting groundwater recharge reaching the groundwater table. The spatial and temporal distribution of this delay determines how fast and how significantly groundwater is affected after periods of meteorological drought. This delay mainly occurs in the unsaturated soil zone of the hydrological cycle. However, studies of drought propagation [4–7] in the hydrological cycle often give less attention to lag effects caused by the vadose zone. Additionally, this effect is often not considered in groundwater modeling, analysis of climate change's impact on groundwater, and effective management and sustainability of future water resources.

Estimation of the time for groundwater recharge to reach the groundwater table needs critical understanding and estimating of hydraulic properties and flow in the vadose zone. Therefore, unsaturated soil flow equations and modeling can be used to estimate this delay. This flow is controlled by the spatial and temporal heterogeneity of the vadose zone and hydrological perturbations at the surface and within the subsurface.

The groundwater recharge delay is mainly influenced by changes in soil moisture and pressure in the vadose zone. These factors are a result of small hydrological perturbations at the surface [8]. Kinematic wave approximation theory can be used to predict these small changes of pressure in the vadose zone [8–14]. According to Lighthill and Whitham [15], a one-dimensional pressure wave in the hydrological system can be described using kinematic wave theory. Therefore, soil moisture content, unsaturated hydraulic conductivity, and vertical soil moisture velocity in the vadose zone can be described using this theory. However, hydraulic properties of the vadose zone should be defined continuously over the spatial and temporal scale of the working domain [11].

This study considers one-dimensional vertical soil moisture movement in the unsaturated soil zone. This vertical flow is governed by capillary and gravity forces. Gravity dominant flow can be applied in the case of deep drainage in response to rainfall and soil with a larger soil moisture content than field capacity [14]. For gravity dominated flow, kinematic wave velocity is used to analyze the vertical soil moisture movement of small changes in pressure perturbations in the unsaturated soil zone [8]. This means that this velocity controls the travel time of pressure distributions in the unsaturated soil zone [12].

Numerical simulations are used for the quantification of small changes in pressure distribution in the vadose zone in the time and space scale. Some studies have used numerical simulations of the vadose zone by solving the 3D Richards equation to determine the travel time of soil moisture [16,17]. However, numerical simulation using the 3D equation leads to complex models and model calibration issues and needs long calculation times as well as extensive data. Therefore, the 1D gravity-driven kinematic wave approximation approach is used for a simplified treatment of flow processes in the vadose zone [18,19].

Estimation of the groundwater recharge delay in the vadose zone can be handled at a point or spatially distributed scale. The studies of Hocking and Kelly [20] and Mattern and Vanclooster [21] estimate the groundwater recharge delay in the vadose zone at point scale. However, other studies such as Rossman et al. [19] estimate lag time using a kinematic wave approximation of the Richards equation in a spatially distributed way, based on the van Genuchten–Mualem parametric model. Rossman et al. [19] mainly focus on the effect of the vadose zone thickness and climate change on the groundwater recharge delay.

This paper aims to estimate the spatially distributed vadose zone lag time in relation to groundwater drought using the kinematic wave approximation of the Richards equation under the assumption of deep drainage occurring in response to rainfall infiltration, steady vertical flow (no flow barriers or artificial drainage), and isotropic and homogenous soil hydraulic properties. This theory is combined with the van Genuchten–Burdine and Brooks–Corey parametric models. Besides the above, the paper will propose a simplified approach for lag time estimation in the vadose zone.

## 2. Study Site and Data

The study area consists of the Dijle and Demer catchments in Central Belgium, underlain by the Brulandtkrijt and Central Campine groundwater systems (Figure 1). The aquifer has a surface area of 5800 km². The land surface elevation ranges between 10 m and 100 m. Belgium has a temperate climate with a long-term average annual rainfall of around 800 mm and an average temperature of around 10 °C [3]. This high and frequent rainfall is the main factor initiating subsurface drainage in the study area. The vadose zone in the study area vertically extends from the Quaternary and Campine aquifer systems (HCOV 0100 & 0200) to the Cretaceous aquifer system (HCOV 1100) shown in Figure 2. However, more than 65 percent of the vadose zone contains geological layers from the Quaternary and Campine aquifer systems. The Quaternary deposits are mostly sandy in the northern part of the study area. In contrast, loamy deposits characterize the hilly regions in the south. Moreover, the central parts of the river valleys (Dijle, Demer, and smaller tributaries) are covered with alluvial deposits, typically loamy or clayey [22].

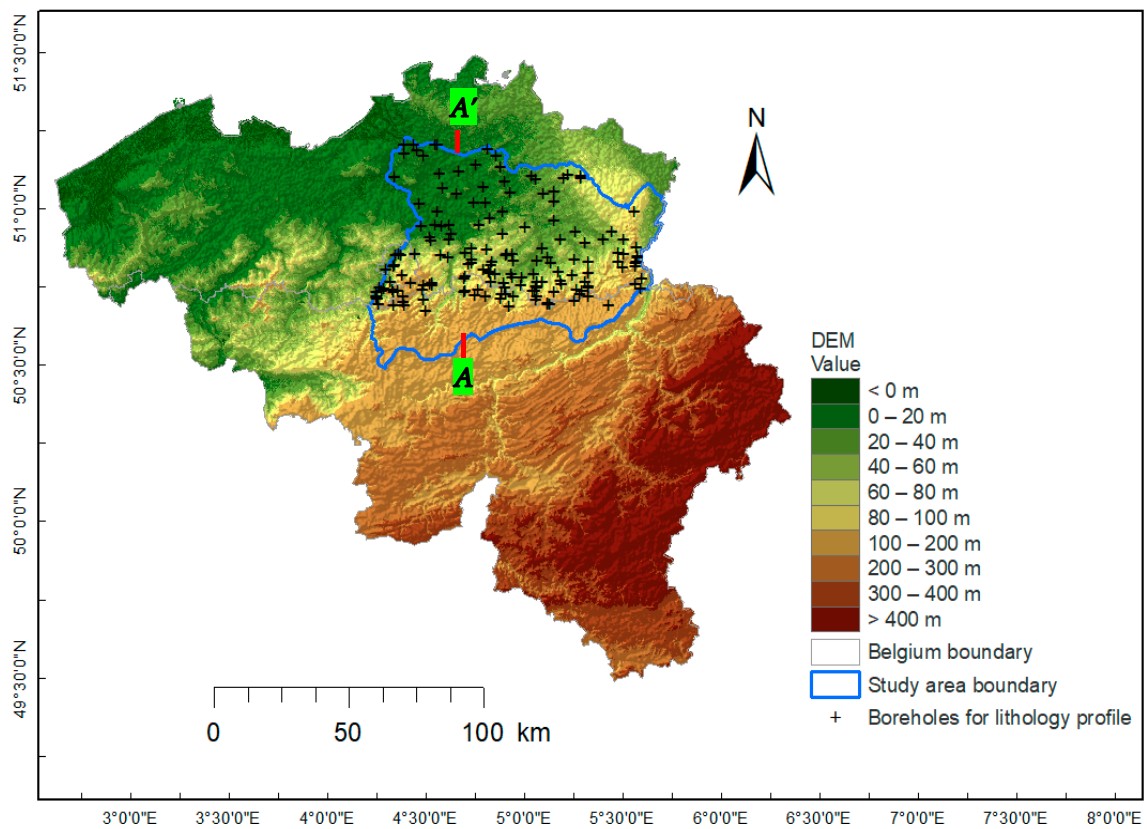

**Figure 1.** Location of the study area, digital elevation model (DEM), and lithology profile boreholes in Belgium.

HCOV is a hydrogeological code developed within the framework of the Flemish groundwater model [23], giving a unique code to every aquifer and aquitard in Flanders.

Geological, hydrogeological, and groundwater data such as borehole information on geology, hydraulic properties, groundwater monitoring wells, etc., is available from the DOV database (Databank Ondergrond Vlaanderen—dov.vlaanderen.be), the Flemish groundwater model [24,25], and the Service Public Wallonie (SPW). Detailed characterization of soil texture, i.e., proportions of sand, silt, and clay, in the vadose zone was taken from 168 soil boreholes (Figure 1). Soil physical properties such as porosity $n$, saturated hydraulic conductivity $K_s$, and pore size distribution index $\lambda$ are dependent on soil texture [26]. The USDA soil texture classification and the univariate regression equation by Cosby, Hornberger, Clapp, and Ginn [27] were used for deriving soil parameter values.

**Table 1.** Geological layers found in the vadose zone of the study area.

| Layer | Groundwater Unit | HCOV Codes | Range of Layer Thickness (m) |
|---|---|---|---|
| 1 | Top of the Quaternary aquifer systems Local deposits of the Campine aquifer system | HCOV 0100 HCOV 0200 | 0–87 |
| 2 | Deposits of the Boom aquitard | HCOV 0300 | 0–14.5 |
| 3 | Oligocene aquifer system | HCOV 0400 | 0–32.5 |
| 4 | Bartoon aquitard system | HCOV 0500 | 0–6 |
| 5 | Ledo Panselian Brusselean aquifer system | HCOV 0600 | 0–35 |
| 6 | Ypresian aquifer system Ypresian aquitard system | HCOV 0800 HCOV 0900 | 0–36 |
| 7 | Paleocene aquifer system | HCOV 1000 | 0–24 |
| 8 | Cretaceous aquifer system | HCOV 1100 | 0–29 |

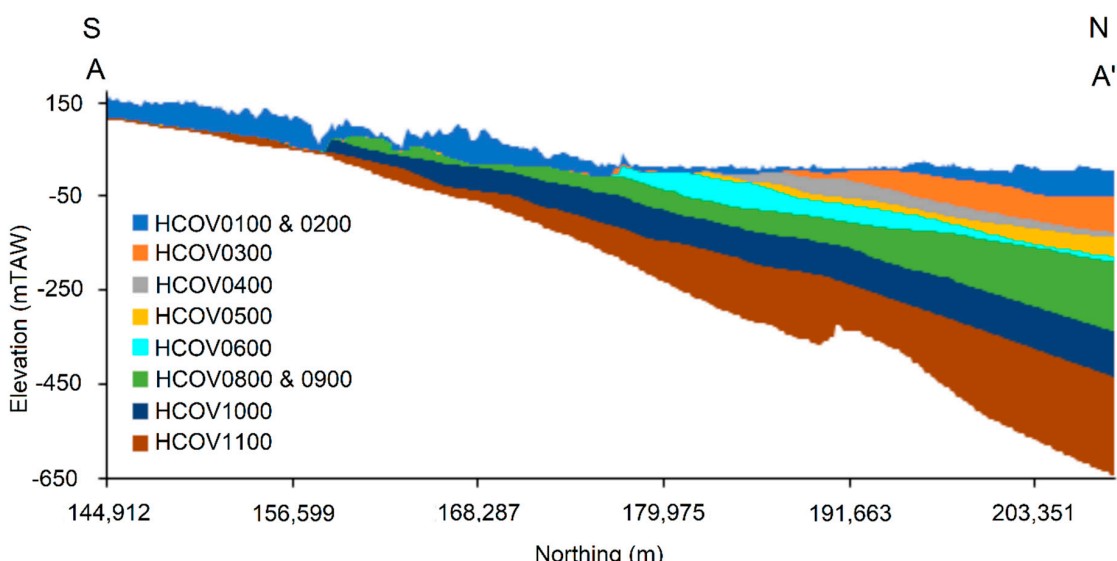

**Figure 2.** Hydrogeological S-N cross-section through the study area. The location of the cross-section is indicated in Figure 1 as AA'. Descriptions of HCOV codes can be found in Table 1. The elevation is measured from a Belgian referencing system for elevation called TAW ("Tweede Algemene Waterpassing").

The vadose zone thickness was estimated as the difference between land surface elevation and groundwater level (Figure 3). A groundwater table map was generated based on monthly groundwater level readings from 1980 to 2013 in observation wells found in the study area. The northern part of the study area has a shallow groundwater table (less than 10 m), while a thicker vadose zone characterizes the southern part of the study area.

Spatially distributed groundwater recharge ($R(t)$) generated in the GroWaDRISK project was used. This project was aiming for the development of a drought-related vulnerability and risk assessment strategy for sustainable management of groundwater resources under temperate conditions [3]. In this project, a monthly averaged, spatially distributed groundwater recharge was estimated using the WetSpaSS model from 1980 to 2013. WetSpaSS is a physically based model for the estimation of spatially distributed surface runoff, actual evapotranspiration, and groundwater recharge [28]. This model accounts for spatially distributed land use, soil type, slope, elevation, monthly average groundwater depth, and meteorological conditions as an input, and it calculates the water balance up to the root zone [29,30]. Therefore, the groundwater recharge computed using this model has to pass through the vadose zone to reach the groundwater table. Figure 4 shows that groundwater recharge is higher than

300 mm/year in the north, which is characterized by sandy soil and gentle slopes. Moreover, in river valleys characterized by loamy soil and steep slopes, groundwater recharge ranges between 100 and 300 mm/year. In urbanized areas, groundwater recharge is often lower than 100 mm/year because of impervious surfaces. In general, the estimated groundwater recharge is larger than 150 mm/year in most of the study area. These high rates of groundwater recharge, along with the presence of the thick permeable aquifers, make the area valuable for its groundwater reserves and drinking water supply.

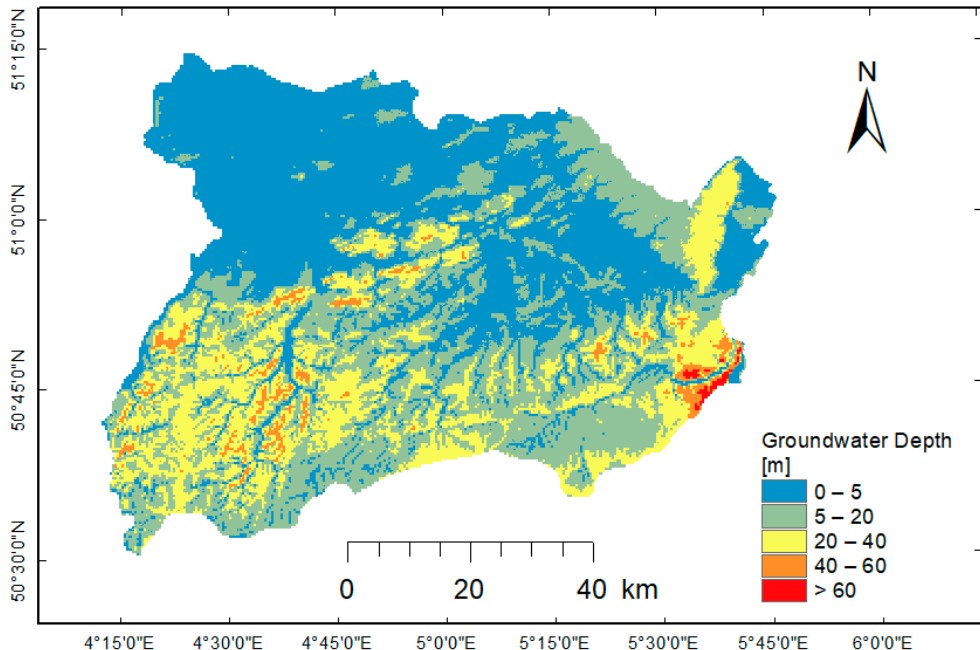

**Figure 3.** Thickness of the vadose zone (m) in the study area, estimated as the difference between 100-m digital elevation model data and averaged water table data (obtained from GroWaDRISK project).

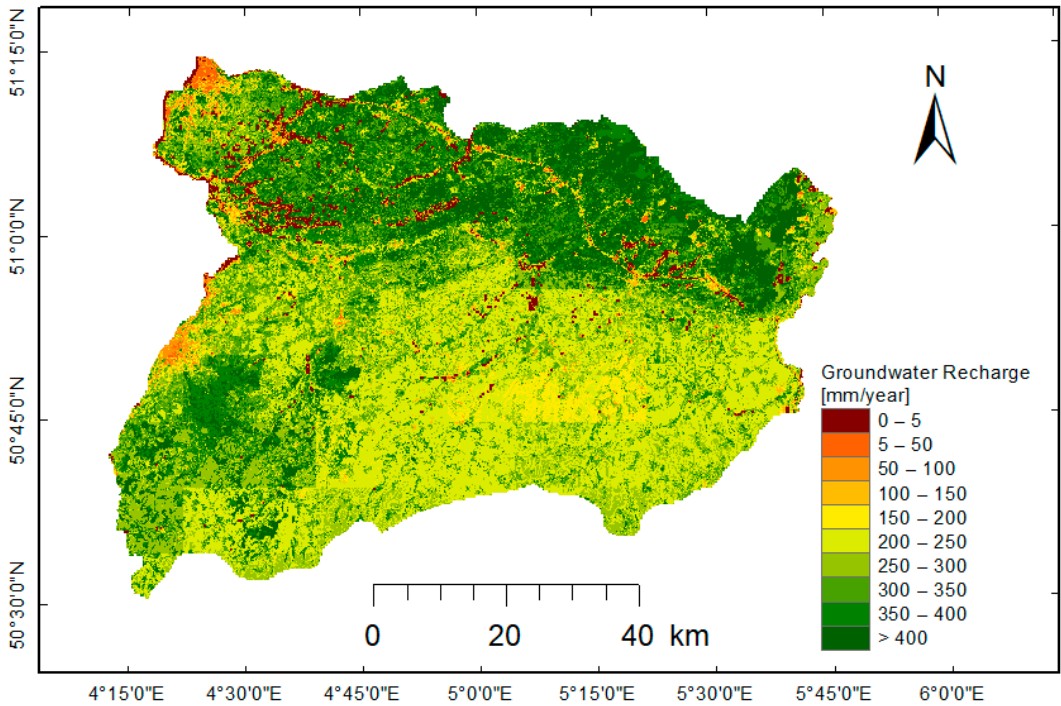

**Figure 4.** Yearly averaged groundwater recharge from 1980 to 2013, estimated using WetSpaSS (the result of GroWaDRISK project).

## 3. Mathematical Model Development

### 3.1. Kinematic Wave Approximation Theories

In this section, the derivation of kinematic velocity from the kinematic wave approximation model in the vadose zone will be presented. The kinematic wave model is primarily based on the continuity equation and only approximates the dynamic equation with uniform flow [31]. As stated in Lighthill and Whitham [15], kinematic waves only propagate in the vertical direction; thus, kinematic waves have one wave speed, which is called celerity or kinematic velocity. The one-dimensional flow of the kinematic wave model in the vadose zone is a functional relationship between the flow (q) and head (h) or soil moisture content (θ).

Smith [14] stated that if the steady gravity-driven vertical porous media flow passing down through the vadose zone, the kinematic wave velocity or celerity can be expressed as the following:

$$c = \frac{dq}{d\theta}$$ (1)

In porous media flow, celerity can be expressed in terms of soil moisture (θ) and unsaturated hydraulic conductivity. Therefore, the Buckingham–Darcy flux law under the assumption of gravity dominant unsaturated flow [32] was used to express this celerity.

The flow in the vadose zone of soil is often assumed as one-dimensional steady gravity dominant flow because the hydrological changes at the surface are often minimal, and subsurface drainage mainly occurs under frequent rainfall. For gravity dominant flow in the vadose zone, with water flowing downward at a constant rate, the matric potential gradient (dh/dz) approaches zero and water flows under the influence of gravity alone. Therefore, the Buckingham–Darcy flux law for gravity-driven flow can be approximated as follows:

$$q \approx -K(h) \approx -K(\theta)$$ (2)

Since unsaturated hydraulic conductivity K(h) is a function of the matric potential head (h), and K(θ) is a function of soil moisture content (θ), K may be written directly as a function of θ.

Celerity or kinematic wave velocity in the vadose zone for steady gravity dominate flow can be derived based on Equations (1) and (2).

$$c = \frac{dK(\theta)}{d\theta}$$ (3)

Equation (3) is used to indicate the timing of a small change in soil moisture or capillary pressure at a water table resulting from a small change of hydrological conditions at the land surface. This velocity is most relevant when calculating the delay of groundwater recharge in the vadose zone [19].

### 3.2. Parameterization of the Kinematic Wave Model

For conditions where the moisture flux is only a function of the moisture content, kinematic wave models are applied to describe the moisture flow behavior in the vadose zone [10]. Unsaturated hydraulic conductivity, porosity, residual soil moisture, and pore size distributions of the vadose zone are used to parametrize the soil moisture content and celerity. In this study, the relationships formulated by Brooks–Corey and van Genuchten–Burdine are used to express soil moisture and celerity(c) in a parametric form based on their unsaturated hydraulic conductivity function.

#### 3.2.1. Unsaturated Hydraulic Conductivity Function

Water flow and solute transport in the subsurface are strongly influenced by hydraulic conductivity. Parameters of the vadose zone, like the pore size distribution of the medium and the tortuosity, shape, roughness, and degree of interconnectedness of the pores, affect the value of hydraulic conductivity in the subsurface [33].

In this study, the closed-form analytical expression based on Burdine's theory for predicting unsaturated hydraulic conductivities is used. Since the Burdine theory is applicable for isotropic media [34,35], the flow is assumed to be in isotropic media. The following equation was derived by Burdine [36] for predicting relative hydraulic conductivity from pore size distribution data and basic laws of fluid flow in a porous medium.

$$K_r = S_e^2 \int_0^{S_e} \frac{1}{h^2(x)} dx \bigg/ \int_0^1 \frac{1}{h^2(x)} dx \tag{4}$$

where h is the pressure head, which is a function of relative soil moisture content or effective saturation, $S_e$:

$$S_e = \frac{\theta - \theta_r}{\theta_s - \theta_r} \tag{5}$$

In Equation (5), $\theta_s$ and $\theta_r$ indicate the saturation and residual soil moisture content in the vadose zone, respectively. Both Brooks–Corey and van Genuchten express the relative soil moisture content in the form of pressure head to solve the Burdine relative hydraulic conductivity equation.

Based on the observation of a large number of experimental data, Brooks and Corey [34,35] express effective saturation in the function of pressure head as follows:

$$S_e = \left(\frac{h}{h_b}\right)^{-\lambda} \tag{6}$$

where $h_b$ is the bubbling pressure (approximately equal to the air entry value), and λ is a number to characterize pore size distribution. Then, the closed-form expression of relative hydraulic conductivity can be derived by solving Equation (4) using Equation (6). Therefore, relative hydraulic conductivity using Brooks–Corey is as follows:

$$K_r = S_e^{(3\lambda+2)/\lambda} \tag{7}$$

van Genuchten [37] also expresses the effective saturation in the function of pressure head:

$$S_e = \left[\frac{1}{1 + |\alpha h|^n}\right]^m \tag{8}$$

where α and n are van Genuchten empirical coefficients which depend on the air bubbling pressure and pore size distribution, respectively, m = 1 − 2/n (with n > 2), and h is the pressure head, assumed to be positive below the root zone or during deep drainage. Then, a closed-form expression for van Genuchten relative hydraulic conductivity can be derived by solving Equation (4) of Burdine theory using Equation (8):

$$K_r = S_e^2 \left[1 - \left(1 - S_e^{\frac{1}{m}}\right)^m\right] \tag{9}$$

Relative hydraulic conductivity in Equations (7) and (9) is equal to the ratio of unsaturated $K(\theta)$ and saturated hydraulic conductivity $K_{sat}$.

$$K_r = K(\theta)/K_{sat} \tag{10}$$

Using relative hydraulic conductivity Equations (7), (9), and (10), parametric models can be developed to estimate unsaturated hydraulic conductivity, $K(\theta)$, from vadose zone properties. Therefore, unsaturated hydraulic conductivity for the two parametric models can be expressed as follows:

Brooks–Corey model (BC):

$$K(\theta) = K_s(S_e)^{\frac{3\lambda+2}{\lambda}} \tag{11}$$

van Genuchten–Burdine model (VGB):

$$K(\theta) = K_s S_e{}^2\left[1 - \left(1 - S_e{}^{\frac{1}{m}}\right)^m\right] \tag{12}$$

According to the soil retention curve resulting from van Genuchten [37], the above two parametric models become identical for a sufficiently low value of soil moisture, $\theta$, under the same parameter values of $\alpha$ and n and $\lambda$ (which is equal to n $-$ 1) [38].

### 3.2.2. Soil Moisture Content

Due to the relatively large size of the catchment, the difficulty of measuring soil moisture content values in the subsurface, and its variability in space and time, soil moisture content was estimated using the Brooks–Corey and van Genuchten–Burdine parametric models.

During the soil moisture transport in the vadose zone, when the moisture content reaches field capacity, hydraulic conductivity is sufficient to pass the volumetric flux q in the downward direction under the force of gravity, so that water content $\theta$ can be inferred from knowledge of q and $K(\theta)$ [39]. The assumption hereby is that under these conditions, the hydraulic gradient is equal to one (flow is only driven by gravity). Hence, the water content of the vadose zone must be the water content at which $K(\theta) = q$. By setting $K(\theta) = q$ in the Brooks–Corey model (Equation (11)) and rearranging, the average soil moisture content in the vadose zone can be calculated as follows:

$$\theta = \theta_r + (\theta_s + \theta_r)\left(\frac{q}{K_s}\right)^{\frac{\lambda}{3\lambda+2}} \tag{13}$$

By similarly setting $K(\theta) = q$ in the van Genuchten–Burdine equation (Equation (12)), the average soil moisture content can be calculated from the recharge flux as follows:

$$\frac{q}{K_s} = S_e{}^2\left[1 - \left(1 - S_e{}^{\frac{1}{m}}\right)^m\right] \tag{14}$$

### 3.2.3. Kinematic Wave Velocity or Celerity

A parametric expression of celerity can be obtained by derivation of the unsaturated hydraulic conductivity formulation of Brooks–Corey and van Genuchten–Burdine with respect to the soil moisture content in the vadose zone.

According to Equation (3), the celerity for the Brooks–Corey equation is obtained by taking the derivative of unsaturated hydraulic conductivity (Equation (11)) with respect to the water content $\theta$, which yields the following:

$$c = \frac{K_s(3\lambda + 2)S_e{}^{\frac{3\lambda+2}{\lambda}}}{\lambda(\theta - \theta_r)} \tag{15}$$

Similarly, the celerity in case of the van Genuchten–Burdine equation is obtained by taking the derivative of Equation (12) with respect to the water content $\theta$, which yields the following:

$$c = \frac{K_s S_e}{(\theta_S - \theta_r)}\left[2(1 - S_f{}^m) + S_e{}^{\frac{1}{m}}S_f{}^{m-1}\right] \tag{16}$$

where $S_f$ is equal to $1 - S_e{}^{1/m}$.

## 4. Methods

In this study, groundwater recharge is assumed as the only vertical water flow in the vadose zone. Thus, groundwater recharging passing through the vadose zone is approximated by the Darcy flow driven by gravity. Therefore, Darcy flux (q) mentioned in all the above equations can be substituted by groundwater recharge.

The parametric models of Brooks–Corey (BC) and van Genuchten–Burdine (VGB) directly address moisture content as the variable of interest and assume gravity as the driving force for water flow. These models were used to estimate soil moisture content and celerity in the vadose zone.

The groundwater recharge delay was calculated from celerity using the following expression:

$$t_d = \frac{T_{vz}}{c} \tag{17}$$

where $t_d$ is the groundwater recharge delay (d) in the unsaturated zone, $T_{vz}$ is the thickness of the vadose zone (m), which is equal to the mean of the groundwater table depth described in Figure 3, and c is the celerity (m/day) calculated with Equation (15) or (16), thereby calculating the moisture content θ with Equation (13) or (14), respectively.

The methods used here to estimate $t_d$ may lead to a small potential error because the capillary fringe near the water table is not taken into account.

### 4.1. Modeling Approaches

In this study, two modeling approaches were applied: (1) using a detailed physical parametrization of the geological layers and (2) using lumped geological layers.

### 4.1.1. Detailed Physical Parametrization of Geological Layers

Detailed soil characterization of the geological layers was taken from 168 soil boreholes found in the Demer and Dijle catchments (Figure 1).

In these soil boreholes, soil texture is aggregated using a soil texture triangle, and the homogeneous soil texture within the thickness of each borehole was considered (Table S1). For each soil texture group, parameter values of physical properties were determined using the USDA soil texture classification and the univariate regression equation by Cosby et al. [27]. Soil moisture content was estimated for each borehole using Brooks–Corey and van Genuchten–Burdine parametric models. After estimating point scale kinematic velocity, empirical Bayesian kriging (EBK) was used to interpolate celerity throughout the study area. During the interpolation of celerity, 80% of the point scale values were used as training data while the rest was used for the validation of interpolation.

### 4.1.2. Lumped Parametrization of Geological Layer

In the second modeling method, the groundwater recharge delay was estimated by lumping the different geological layers found in the vadose zone. The thickness of each geological layer was computed from its top surface elevation, the bottom elevation of each layer, and water table elevation (Figure S1). The data for the elevations of each geological layer were taken from VMM (Vlaamse MilieuMaateschappij) [40]. Hydraulic and physical properties of the geological layers were taken from the hydrogeology study of North-East Belgium by Vandersteen and Gedeon [22].

Spatially distributed, yearly averaged soil moisture content was estimated for each geological layer using the Brooks–Corey and van Genuchten–Burdine parametric models. Moreover, celerity was estimated in each geological layer based on the result of the soil moisture content and yearly averaged groundwater recharge. The groundwater recharge delay in the vadose zone was estimated using the relationship of celerity and vadose zone thickness shown in Equation (17).

### 4.2. Comparison with Other Methods

Direct measurement of the delay time in the unsaturated zone is not possible, and validation is very challenging, especially for the size of the study area. As direct validation of the resulting delay time is not possible, the model results were compared to results from different models [41]. The results

of the above parametric models are compared to a physically based flow model, HYDRUS-1D, which solves the 1-D Richards flow equation numerically.

$$\frac{\partial \theta}{\partial t} = \frac{\partial}{\partial z}\left[K(h)\left(\frac{\partial h}{\partial z} + 1\right)\right] - S \tag{18}$$

where z is the vertical coordinate (positive upward), t is time (d), S is a sink term ($d^{-1}$), h is the pressure head (m), $\theta$ is the volumetric moisture content, and K is the unsaturated hydraulic conductivity function (m/d).

The results of both the van Genuchten–Burdine and Brooks–Corey parametric models were compared to the HYDRUS-1D model for 15 selected locations varying in the thickness of the vadose zone. For these locations, the lithological information could be inferred from borehole data. The soil physical properties were parametrized based on USDA default soil texture parameter values mentioned in Cosby et al. [27].

A variable flux condition was imposed on the soil surface, considering the same inputs R(t) as those used in the parametric models. Since the bottom of the flow domain is the groundwater table, a constant soil moisture content equal to the saturated soil moisture content was imposed. Field capacity soil moisture content throughout the vadose zone was defined as the initial condition. Moreover, the soil hydraulic property model was set as Brooks–Corey and van Genuchten–Mualem for different simulations.

The groundwater recharge delay was calculated by comparing the timing of the peak flow daily time series at the soil surface and the groundwater table.

### 4.3. Delay Effect on Groundwater Drought

To assess the effect of the delay in the vadose zone on groundwater drought, the groundwater recharge delay output was implemented on the spatially distributed, monthly recharge time series between 1980 and 2013 obtained from the GroWaDRISK project, resulting in a delayed recharge time series $R(t)_d$.

A groundwater drought analysis was performed on the original groundwater recharge (R(t)) and the delayed groundwater recharge ($R(t)_d = R(t - t_d)$) time series of 34 years. The threshold level method introduced by Yevjevich [42], with a variable threshold value [43], was used to determine the occurrence of groundwater recharge drought events.

To do so, a separate frequency analysis was conducted for each of the 12 months. The threshold level for each month was determined as the 70th percentile of the probability of exceedance of monthly recharge in that month in the 34-year series (1980–2013). This threshold level is within the 70th–95th percentile of the probability of exceedance range used for most drought studies [44]. For each month of the year, the monthly recharge values in the time series were ranked from highest to lowest. For each month, percentiles were calculated using the formula from [45]:

$$P_i = 100\frac{i - 0.5}{n} \tag{19}$$

where $P_i$ is the percentile of the data set, i is the rank number, and n is the total number of data points. The formula was rearranged to calculate the rank for the 70th percentile recharge for each month, and the corresponding threshold value of recharge was obtained using linear interpolation between successive ranks. Finally, the comparison of drought events of groundwater recharge and delayed groundwater recharge were made in terms of the number of drought events, the severity of drought, and their duration and timing. The results were analyzed for the whole study area and also separately for the subareas with deep and shallow groundwater tables, defined as areas with an average groundwater depth higher than 40 m and lower than 10 m, respectively.

## 5. Results and Discussion

### 5.1. Detailed Physical Parametrization of Geological Layers

#### 5.1.1. Soil Moisture and Celerity

Estimated soil moisture content in the study area varies from 0.03 to 0.25 mm/mm (Figure 5). The van Genuchten–Burdine model gives a higher estimation of soil moisture than the Brooks–Corey model. Soils with low hydraulic conductivity and high deep percolation have higher soil moisture content.

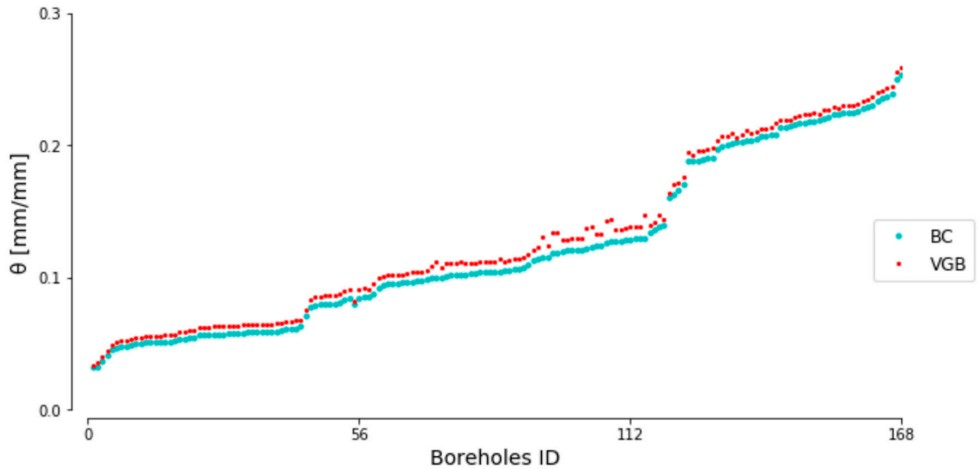

**Figure 5.** Average soil moisture content for 168 locations found in Dijle and Demer catchment, ordered from low to high soil moisture content.

Celerity estimated using the van Genuchten–Burdine model ranges from 179 to 6367 mm/month, while the Brooks–Corey model estimates values from 148 to 3597 mm/month (Figure 6). Due to the different estimates of soil moisture in the two models, the celerity estimated by the Brooks–Corey model has lower values compared to the van Genuchten–Burdine model.

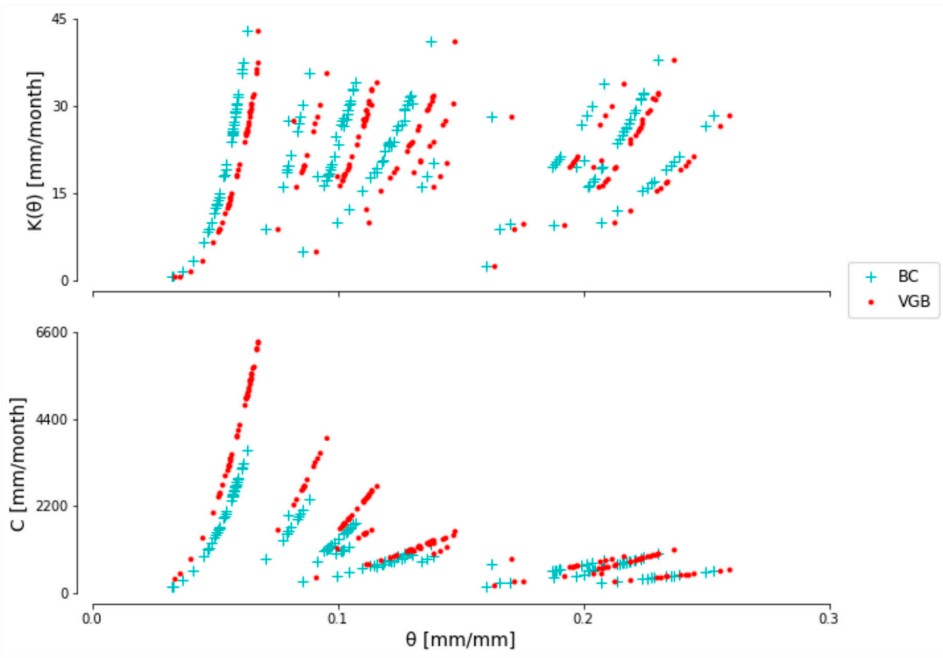

**Figure 6.** Van Genuchten–Burdine and Brooks–Corey model soil moisture content and celerity estimation difference using the same groundwater recharge input for 168 sample boreholes.

The spatially distributed celerity using empirical Bayesian kriging is shown in Figure 7. The performance of the interpolation method is indicated in Table 2.

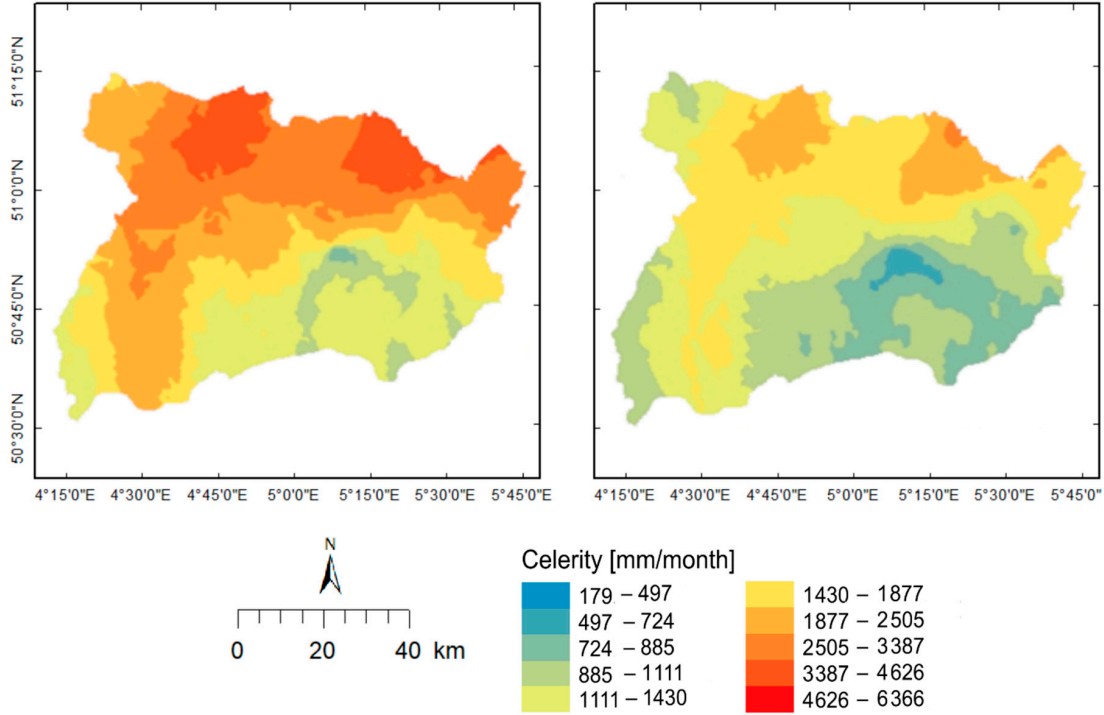

**Figure 7.** Celerity of soil moisture in the vadose zone estimated by van Genuchten–Burdine (**right**) and Brooks–Corey model (**left**).

**Table 2.** Prediction error of empirical Bayesian kriging.

| Prediction Error | VGB | BC |
|---|---|---|
| Root mean square (mm/month) | 1461 | 691 |
| Mean standardized (–) | −0.0109 | −0.0034 |
| Root mean square standardized (–) | 0.994 | 0.989 |
| Average standard error (mm/month) | 1452 | 686 |

5.1.2. Groundwater Recharge Delay

Figure 8 shows the distribution of the groundwater recharge delay. The estimated groundwater recharge delay in the vadose zone using the Brooks–Corey model ranges from 0 to 110 months, with a spatial mean of 10.32 months, whereas, under the van Genuchten–Burdine model, the delay ranges from 0 to 73 months, with a spatial mean of 6.65 months. The difference map shown in Figure 9 shows the spatial distribution of the difference in delay between the Brooks–Corey and van Genuchten–Burdine models. The spatially averaged mean difference is 3.6 months.

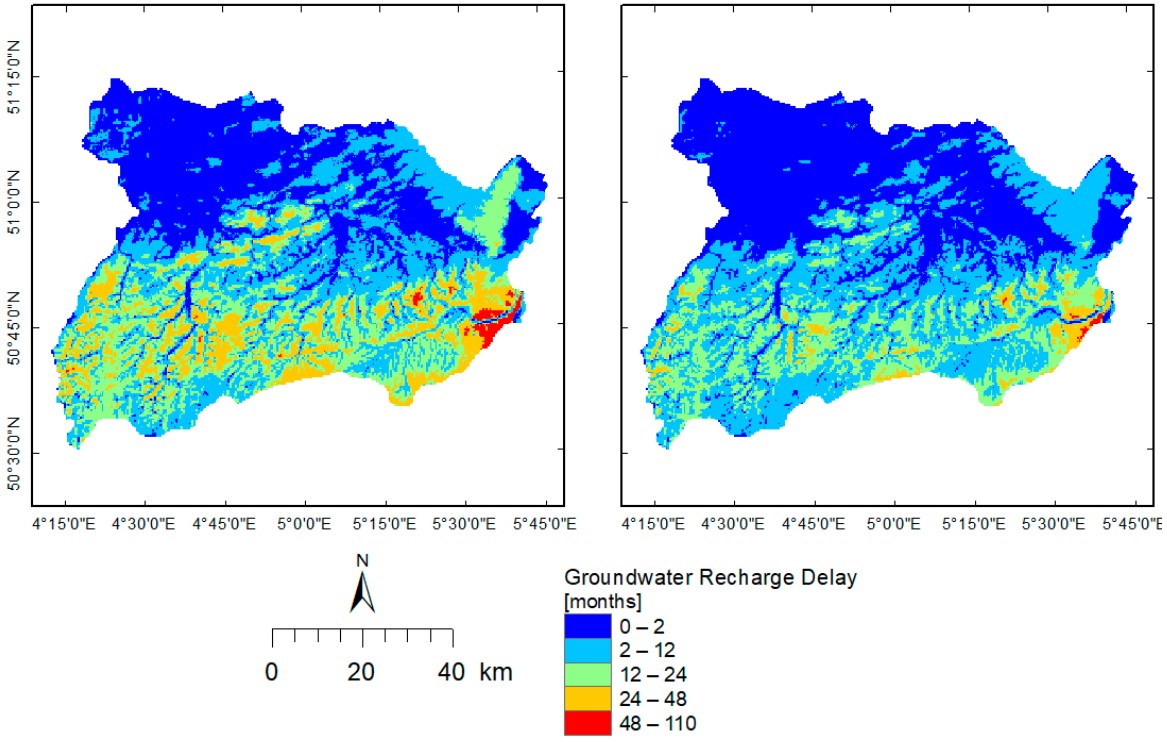

**Figure 8.** Spatially distributed groundwater recharge delay in the vadose zone using the Brooks–Corey parametric model (**right**) and van Genuchten–Burdine parametric model (**left**).

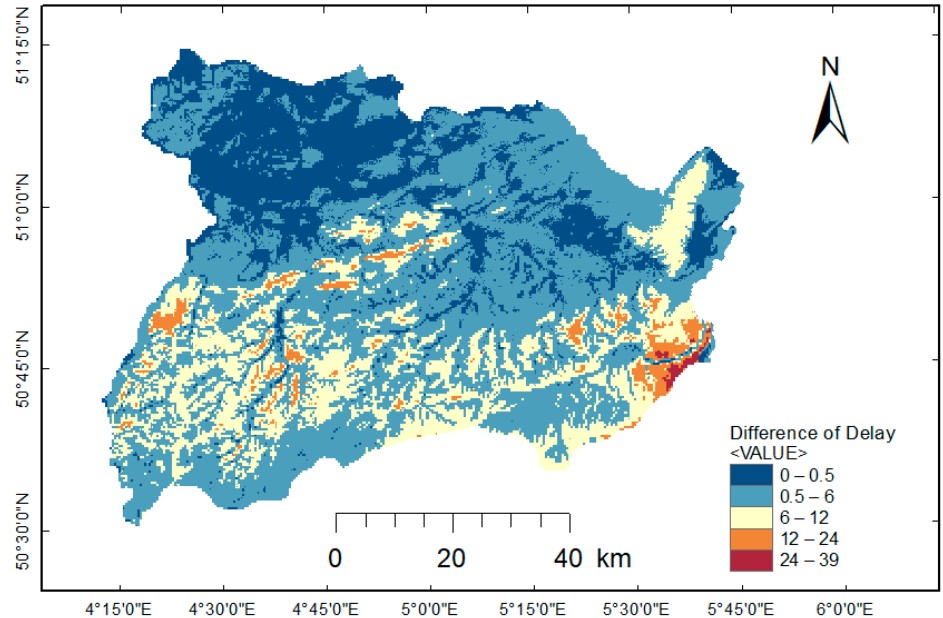

**Figure 9.** Groundwater recharge delay difference between the delay estimated by the Brooks–Corey model and van Genuchten–Burdine model.

Comparing the spatial distribution of the recharge delay and Figure 3, it is clear that the distributions of the vadose zone thickness and the delay have the same pattern. This indicates that the thickness of the vadose zone is the main factor influencing the groundwater recharge delay (Figure 10). In the northern part of the study area with shallow groundwater, the groundwater recharge delay is below 2 months, while the delay increases to 110 months in the area found in the southeast with a deep water table.

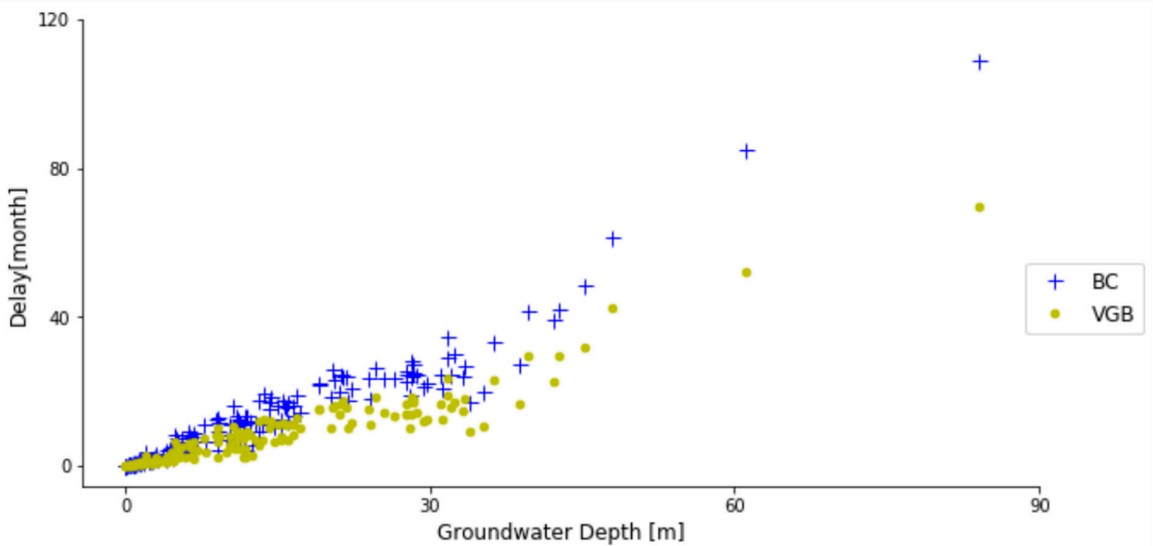

**Figure 10.** Groundwater recharge delay relationship with the thickness of the vadose zone.

*5.2. Lumped Parametrization of Geological Layers*

5.2.1. Soil Moisture and Celerity

The van Genuchten–Burdine and Brooks–Corey models were also used to estimate soil moisture content and celerity for each geological layer found in the vadose zone of the study area. Figure 11 shows that soil moisture and celerity in the vadose zone are mainly influenced by the hydrogeologic properties of the geological layers (mainly saturated hydraulic conductivity) and groundwater recharge. As in the detailed parameterization approach, for a given groundwater recharge, the van Genuchten–Burdine model gives a higher estimation of celerity than the Brooks–Corey model.

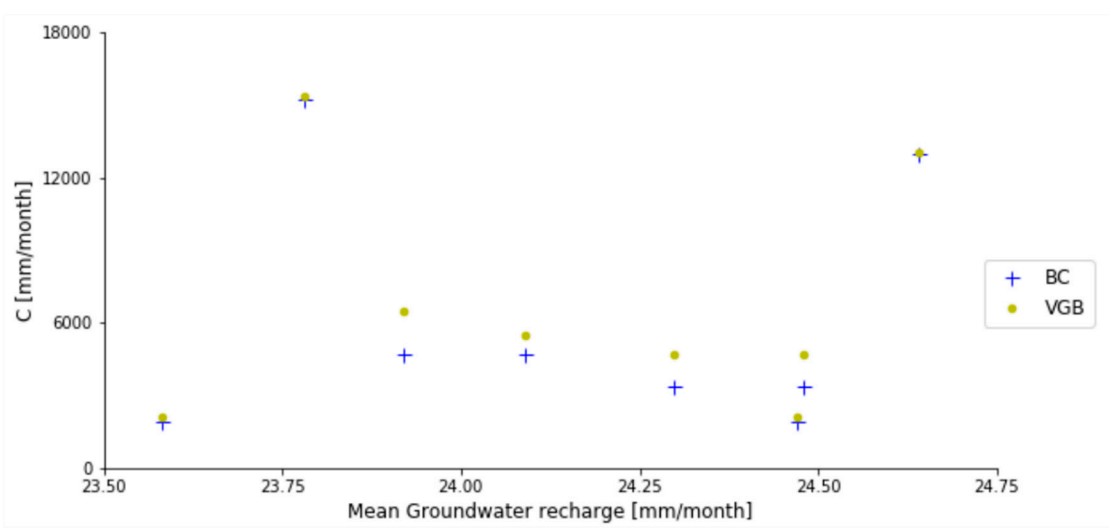

**Figure 11.** Van Genuchten–Burdine and Brooks–Corey model celerity estimation difference using the same groundwater recharge input for eight geological layers found in the vadose zone.

5.2.2. Groundwater Recharge Delay

Figures 12 and 13 show the distribution of the groundwater recharge delay. The estimated groundwater recharge delay in the vadose zone using the Brooks–Corey model ranges from 0 to 26 months. Under the van Genuchten–Burdine model, the delay ranges from 0 to 18 months.

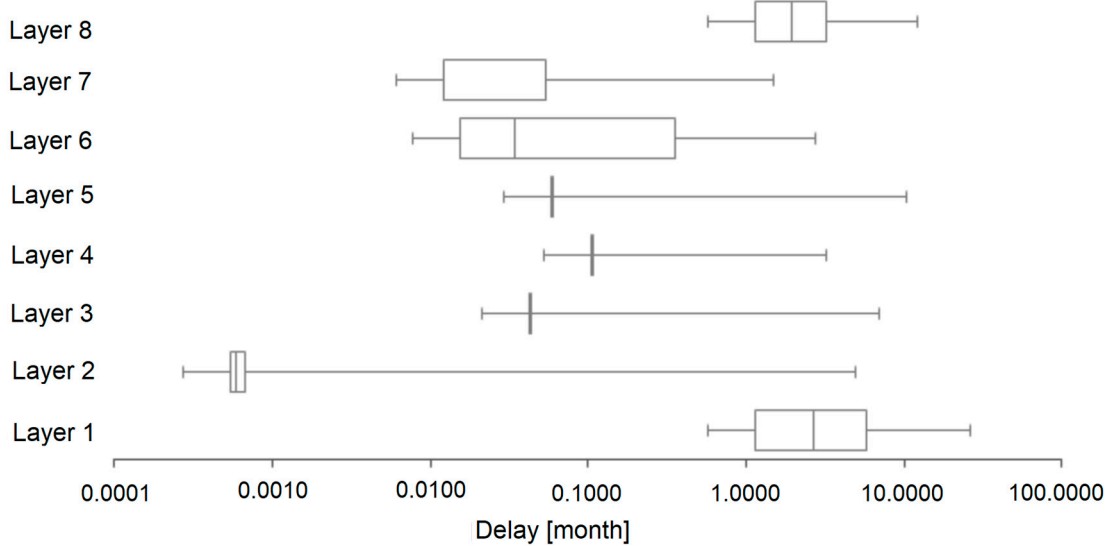

**Figure 12.** Box plots showing the distribution of groundwater recharge delay in each geological layer of the vadose zone using the Brooks–Corey parametric model. See Table 1 for the description of layers.

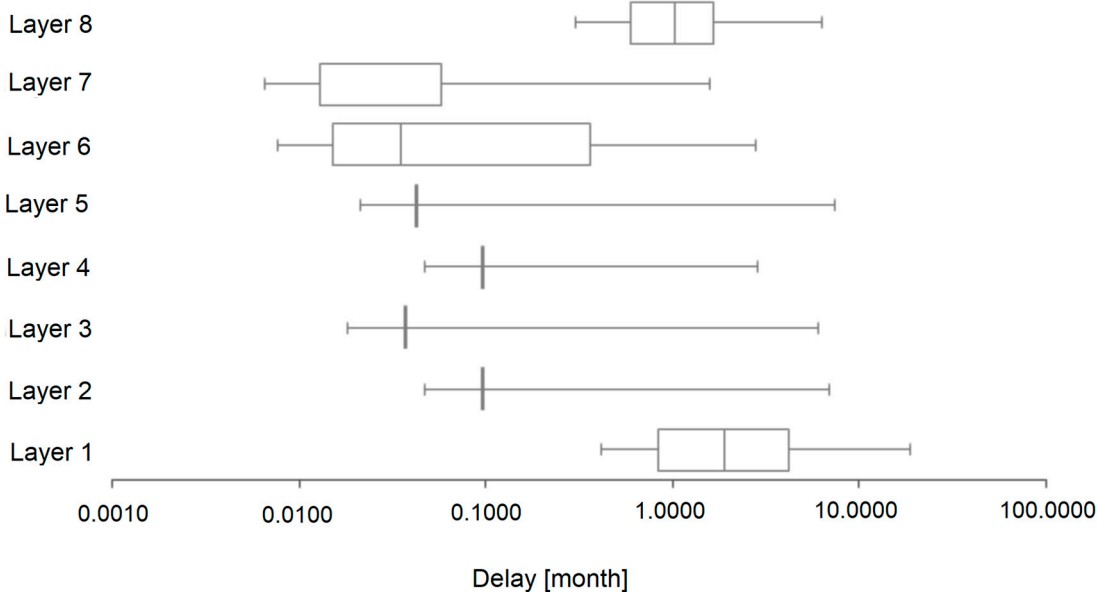

**Figure 13.** Box plots showing the distribution of groundwater recharge delay in each geological layer of the vadose zone using the van Genuchten–Burdine parametric model. See Table 1 for the description of layers.

The comparison of the delay estimated in each geological layer using both models allows examination of the contribution of each layer. As most of the vadose zone is found in Quaternary (HCOV 0100) and deposits of the Kempen (HCOV 0200) aquifer systems, the greatest delay occurs in layer 1.

### 5.3. Comparison of Groundwater Recharge Delay to Physically Based Conceptual Flow Model

The delay estimated using the above parametric models from both modeling approaches was compared to output from a physically based flow model (i.e., HYDRUS-1D). Figure 14 shows that the approach using a detailed parametrization of the geological layers compares better to the HYDRUS-1D output, with a Nash–Sutcliffe efficiency of 0.89 for Brooks–Corey and 0.79 for van Genuchten–Burdine.

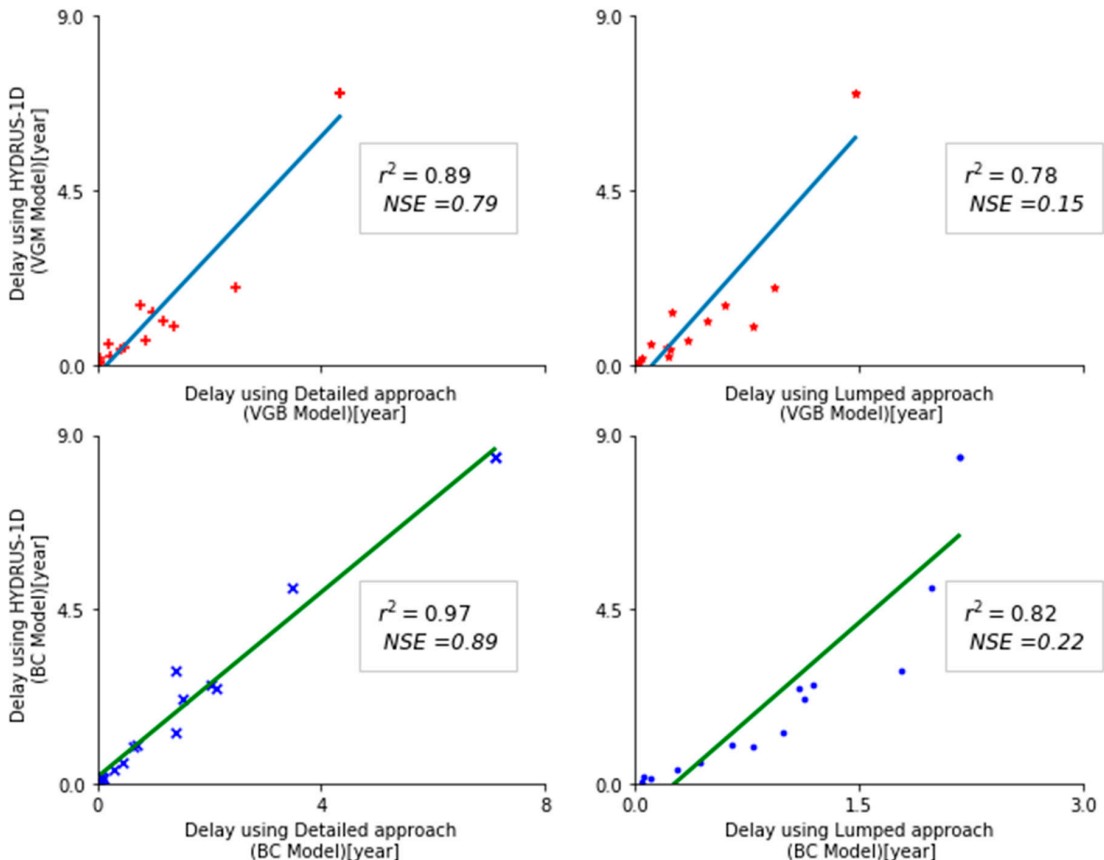

**Figure 14.** Comparison of HYDRUS-1D output with groundwater recharge delay estimated using Brooks–Corey and van Genuchten–Burdine models for both detailed parametrization and lumped approaches.

The models' output from the lumped geological layer approach compares poorly to HYDRUS-1D, with a Nash–Sutcliffe efficiency of 0.22 for Brooks–Corey and 0.15 for van Genuchten–Burdine. However, as a result of the systematical underestimation of the groundwater recharge delay regarding the output of the HYDRUS-1D model, the coefficient of determination ($r^2$) is larger than 75%.

*5.4. Delay Effect on Groundwater Drought*

The effect of the recharge delay on the occurrence and timing of groundwater drought was analyzed using a variable threshold method, where the threshold value was calculated as the 70th percentile of the probability of exceedance for the original and delayed recharge time series, for the whole study area and the subareas with deep and shallow groundwater tables.

5.4.1. Delay Effect on Groundwater Drought for the Whole Study Area

Figure 15 shows the deviation of groundwater recharge from the threshold value in time. Negative anomalies indicate groundwater drought. The results indicate that the delay in the vadose zone decreases the number of drought events from 4 to 10 events and the cumulative deficit volume from 46.87 to 49.51 mm. However, the average duration of drought events increases from 3 to 8 months.

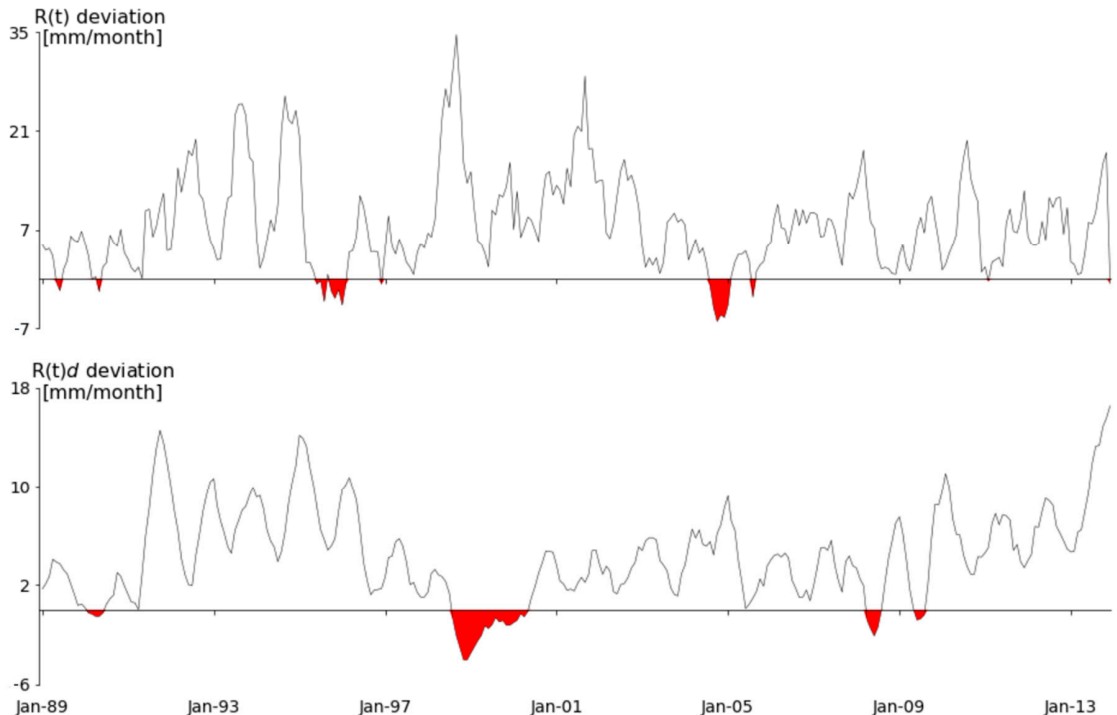

**Figure 15.** Deviation from the threshold value of spatially average monthly groundwater recharge (**top**) and delayed recharge (**bottom**) for the whole study area based on 6-month moving average. The red color represents groundwater drought.

This shows that including the recharge delay in the unsaturated zone results in fewer but longer groundwater drought events, with a lower cumulative deficit volume. Passage of recharge through the unsaturated zone has a smoothing and delaying effect on the occurrence of groundwater drought.

### 5.4.2. Delay Effect on Groundwater Drought for Deep Water Table Area

Monthly groundwater recharge and delayed groundwater recharge for the deep groundwater area with an average groundwater depth of 48 m and delay of 61 months was used to estimate groundwater drought events. Figure 16 shows the deviation of both recharges from the threshold value. The result indicates that the first groundwater drought happened on the delayed groundwater recharge in July 1994, 61 months later than the first drought event in the original groundwater recharge time series. The cumulative deficit of delayed groundwater recharge is 4.8 mm lower. Moreover, considering the delay in the vadose zone, the number of events decreased from 9 to 11 events. This could be because of the larger attenuation in this deep vadose zone and the storage properties of the vadose zone.

The results also show that the average duration of drought events for the delayed recharge and original recharge is 2.6 months and 2.5 months, respectively. Therefore, the delay in areas with deep groundwater tables would mainly affect the onset of drought events.

### 5.4.3. Delay Effect on Groundwater Drought for Shallow Water Table Area

In the shallow groundwater area, a cumulative deficit of 91.56 mm was calculated, versus 92.92 mm for the recharge time series without delay. This small difference in cumulative deficit could be explained by the smaller attenuation in the thin vadose zone and smaller delay time.

The number of drought events estimated using both recharge time series is the same, i.e., ten events. However, Figure 17 shows that the first groundwater drought with delayed recharge happened in Oct. 1983, 3 months after the first drought event in the undelayed groundwater recharge time series.

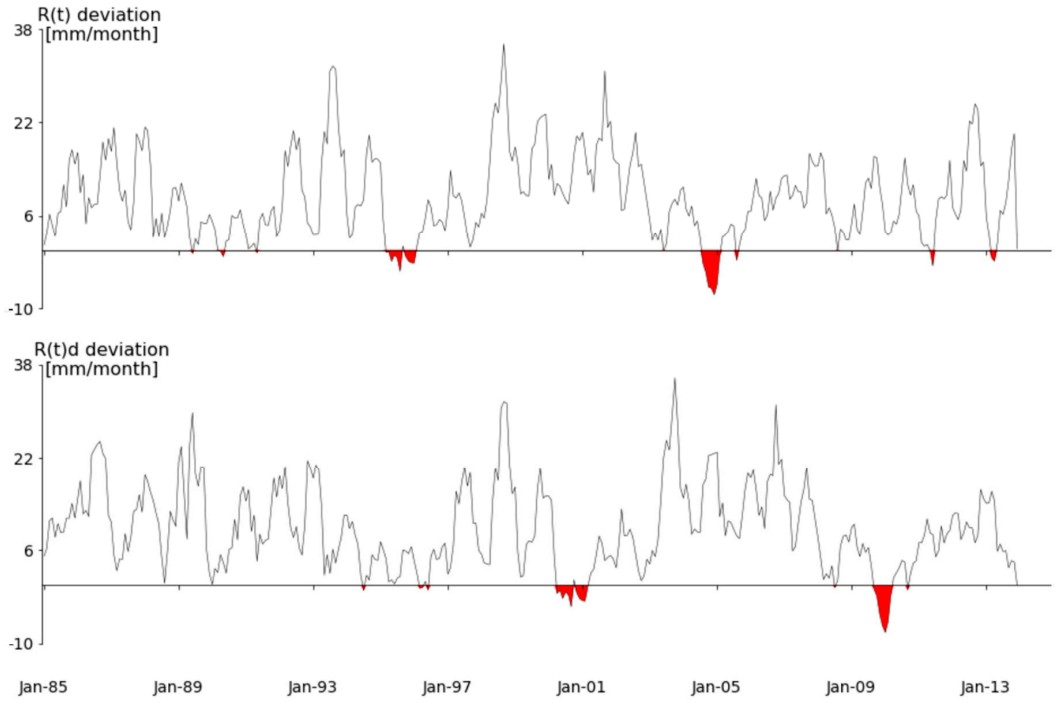

**Figure 16.** Deviation from the threshold value of monthly groundwater recharge (**top**) and delayed groundwater recharge (**bottom**) based on 6-month moving average aggregation. The red color represents a groundwater drought. This drought analysis is for the deep groundwater table area with groundwater depth greater than 48 m.

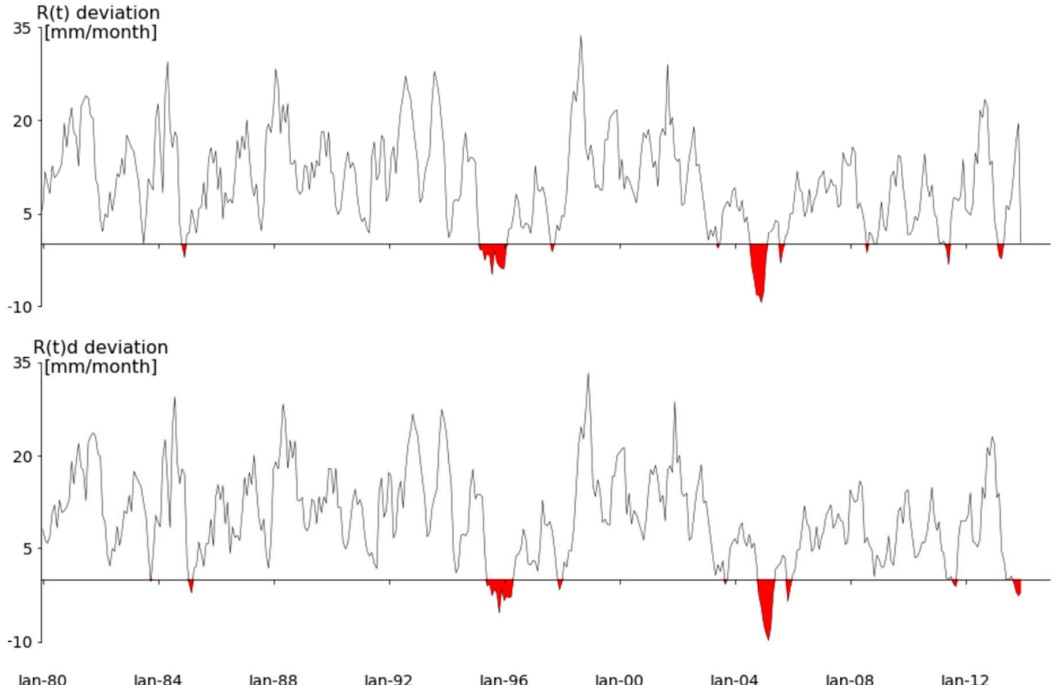

**Figure 17.** Deviation of monthly groundwater recharge (**top**) and delayed groundwater recharge (**bottom**) based on 6-month moving average aggregation from the threshold value. The red color represents a groundwater drought. This drought analysis is for a shallow groundwater table area with maximum groundwater depth of 5 m.

## 6. Conclusions

In this study, spatially distributed groundwater recharge delay through the vadose zone was estimated in the Dijle and Demer catchments in Central Belgium. This estimation was conducted using two mathematical models developed by combining the kinematic wave approximation of Richards' equation with two parametric models describing the soil hydraulic properties, i.e., the Brooks–Corey and van Genuchten–Burdine models.

The vertical soil moisture velocity or celerity was influenced by saturated hydraulic conductivity and soil moisture content in the vadose zone for both models. The Brooks–Corey model estimated lower celerity compared to the van Genuchten–Burdine model.

The calculated delay was compared to the physically based flow model HYDRUS-1D. Delays calculated based on a detailed parametrization of the soil and geological layers are very similar to the HYDRUS-1D results, with a Nash–Sutcliffe efficiency of 0.89 for Brooks–Corey and 0.80 for van Genuchten–Burdine. However, in both models, there is a slightly systematic underestimation of the delay compared to the HYDRUS-1D model.

The estimated spatially distributed groundwater recharge delay in the vadose zone using the Brooks–Corey model ranges from 0 to 110 months, with a spatial mean of 10.3 months. Using the van Genuchten–Burdine model, the delay ranges from 0 to 73 months, with a spatial mean of 6.7 months. The thickness of the vadose zone is the main parameter influencing the spatial distribution of this delay. In regions with shallow groundwater (<10 m), the groundwater recharge delay is less than 2 months, whereas the delay is up to 110 months for areas with a deep water table (>40 m).

The delay effect on groundwater drought was analyzed by implementing the delay result of the Brooks–Corey model on the groundwater recharge time series from 1980 to 2013. Groundwater drought characterization features like number, duration, onset, and intensity of drought events were influenced by the delay in the vadose zone.

Finally, from this study, the Brooks–Corey model with a detailed parametrization of the geological layers is recommended to estimate spatially distributed groundwater delay. Besides this, incorporating this delay in groundwater drought delay analysis does affect the timing of groundwater drought significantly. Therefore, combining the delay in the vadose zone with drought propagation in the hydrological cycle is recommended for future studies. The applicability of the findings and methodology of this research depends on the nature of the vadose zone. In this paper, the method is applied and tested in low land areas. The applicability of the method in mountain catchments needs further investigation.

**Supplementary Materials:** The following are available online at http://www.mdpi.com/2073-4441/12/8/2123/s1, Figure S1: Thickness of geological layers in vadose zone (m) of the study area within 5 color-coded classes, estimated as the difference in 100-m digital elevation model data, base elevation of each geological layers and 34 years (from 1980 to 2013) averaged water table digital data (refer layer type from Table 1 in the manuscript), Table S1 Aggregated soil lithology for 168 soil boreholes.

**Author Contributions:** Conceptualization, B.K.W., B.V., and M.H.; data collection, B.K.W., B.V., and M.H.; mathematical equation development, B.K.W.; methodology, B.K.W., B.V., J.D. and M.H.; supervision, M.H. and B.V.; writing—original draft, B.K.W.; and writing—review and editing, B.K.W., B.V., J.D., and M.H. All authors have read and agreed to the published version of the manuscript.

**Funding:** This research was supported by a KU Leuven Interfaculty Council for Development Cooperation (IRO) PhD scholarship.

**Acknowledgments:** The research idea, spatially distributed groundwater recharge time series, and average groundwater depth data were obtained from the GroWaDRISK project (SD/RI/05a, 2012–2016), funded by the Belgian Science Policy Office (BELSPO) within the framework of the "Science for Sustainable Development" research program. Soil borehole information and geological layer thickness of the study area were obtained from the website of Databank Ondergrond Vlaanderen at https://.dov.vlaanderen.be/, last consulted on May 2017.

**Conflicts of Interest:** The authors declare no conflict of interest.

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
