# Peer review of "Vadose Zone Lag Time Effect on Groundwater Drought in a Temperate Climate"

_water, doi:10.3390/w12082123_

Round 1

Reviewer 1 Report

The manuscript mainly focuses on a topic about effect on groundwater drought for vadose zone lag time. In the text, the spatially distributed groundwater recharge delay through the vadose zone was estimated in a typical catchment in central Belgium. The estimation was done using two mathematical models developed by combining the kinematic wave approximation of Richards’ equation. I am not well familiar about the detail numerical simulations. I think it looks like an interesting theoretical issue. Thus, I recommend it to publish in Water.

This study simply considers one-dimensional vertical soil moisture movement in the unsaturated soil zone. It’s true that the numerical simulation using the 3D equation leads to complex models and model calibration issues and needs long calculation times as well as extensive data. However, as my observation, there rarely appears one-dimensional vertical soil moisture movement dominated water-vadose system in the nature condition. So, the practical utilization of some results or indexes concluded in this study seems to be difficult to assess. Moreover, authors didn’t measure the real parameters such as important soil moistures and only depend on empirical records. How to evaluate on the simulated results are accurate or not.

Author Response

Dear Reviewer,

Best regards,

Authors

Reviewer 2 Report

The article presents an interesting approach to assessing the lag time effect on groundwater drought, based on studies carried out in central Belgium. It fits well to the problem of a groundwater resources decrease due to the ongoing drought in many regions of the world. The results are clearly descriped, but I would expect a short supplement to the description of the lithology of the vadose zone, mainly in Quaternary deposits which play the  major role in the process of groundwater supply.

The calculations were carried out for a relatively large area, so they had the character of spatial regional studies, but please comment, wether proposed estimations  would be equally effective in mountain catchments as in lowlands?

other remarks:

  • line 96: km2 instead of Km2
  • in fig. 1: explain the abbreviation DEM  (hypsometry of Belgium?)
  • in fig. 1: instead of soil texture profile...I suggests to insert lithology profile. The term soil refers mainly to the upper, sub-surface parts of the unsatutaed zone;
  • there is no need to insert N direction on north-oriented maps,
  • fig. 2: explain the abbreviation TAW (an elevation  is given in meters above sea level - m a.s.l.),
  • Soil moisture content was calculated using Brooks-Corey and van Genuchten-Burdin equation, but whether the authors measured the actual soil moisture of the vadose zone in some boreholes to verify the calculations?
  • line 269: is better to write taken into account,
  • improve the quality of fig. 6

Author Response

Dear Reviewer,

Best regards,

Authors

Reviewer 3 Report

I carefully read the manuscript water-843946 untitled “Vadose Zone Lag Time Effect on Groundwater Drought in a Temperate Climate”. This is a well-written manuscript highlighting an important research topic worth of further investigation and can be very useful among the geoscience community. I find the manuscript well structure, interesting to read. The abstract is explicative, keywords are appropriate and introduction is well written. The research method and data analysis techniques follow well-established routines and complying with state-of-the-art methodologies. I felt the figures, result sections need further improvement for better clarification.

I recommend for publication after addressing the following issues:

  1. Cross-section area AA’ is missing in Figure 1 but mentioned in Figure 2.
  2. Latitude and longitude are missing in all the figures. Make no sense to include the measurement scale if latitude and longitude don’t mention on the figures.
  3. Comparing Figure 1, Figure 3 and Figure 4, could authors elaborate why the groundwater recharge maximum in high elevation region with greater groundwater depth?
  4. The title highlights the groundwater drought but authors don’t discuss about the drought indicator over the regions. I think its essential clarifying or showing evidence on groundwater drought over the region and the causes of such effect.
  5. Authors discus about the groundwater recharge in the region but no further discussion on the groundwater discharge during the observation period.
  6. Does the region experience any surface deformation or uplift due to annual variation of aquifer-system in the region?
  7. Results validation are missing using any geodetic measurements. Could authors discuss on the validation of their results and elaborate on discussion section.

Author Response

Dear Reviewer,

Best regards,

Authors

Round 2

Reviewer 3 Report

I appreciate authors comprehend response to my comments. I recommend for publication with minor spell check required.